# Positioning Accuracy in Holographic Optical Traps

**DOI:** 10.3390/mi12050559

**Published:** 2021-05-15

**Authors:** Frederic Català-Castro, Estela Martín-Badosa

**Affiliations:** 1Optical Trapping Lab, Grup de Biofotònica (BiOPT), Departament de Física Aplicada, Universitat de Barcelona, 08028 Barcelona, Spain; frederic.catala@icfo.eu; 2Institut de Nanociència i Nanotecnologia (IN²UB), 08028 Barcelona, Spain

**Keywords:** spatial light modulators, laser trapping, holographic optical tweezers

## Abstract

Spatial light modulators (SLMs) have been widely used to achieve dynamic control of optical traps. Often, holographic optical tweezers have been presumed to provide nanometer or sub-nanometer positioning accuracy. It is known that some features concerning the digitalized structure of SLMs cause a loss in steering efficiency of the optical trap, but their effect on trap positioning accuracy has been scarcely analyzed. On the one hand, the SLM look-up-table, which we found to depend on laser power, produces positioning deviations when the trap is moved at the micron scale. On the other hand, phase quantization, which makes linear phase gratings become phase staircase profiles, leads to unexpected local errors in the steering angle. We have tracked optically trapped microspheres with sub-nanometer accuracy to study the effects on trap positioning, which can be as high as 2 nm in certain cases. We have also implemented a correction strategy that enabled the reduction of errors down to 0.3 nm.

## 1. Introduction

The possibilities provided by optical tweezers in the manipulation of specimens at the micron and sub-micron scale have been widely investigated since their discovery in 1970 [1,2]. The use of devices such as acousto-optic deflectors (AODs) [3,4] or galvano mirrors [5] has allowed dynamic positioning of optical traps. Compared to these solutions, holographic optical tweezers (HOT) [6,7] offer useful and novel capabilities derived from beam shape engineering: creation of multiple optical traps in 3D [8,9,10], aberration compensation [11] and generation of exotic laser beams with complex optical trapping potentials [12,13].

In particular, the use of spatial light modulators (SLMs) for wavefront shaping can be applied to accurate beam steering and, therefore, to precise positioning of optical traps [8,14,15]. Commonly, phase-only spatial light modulators are used and a single beam is easily deflected by generating phase gratings. The digital structure and the phase quantization of the SLM limit the resolution in beam positioning but, for commercial SLMs with *N*∼500 pixels and *M*∼200 phase values, theoretically, almost continuous placement should be achieved [8]. In [16], a detailed analysis of the effects of a 1D phase only SLM on beam steering is performed. They show that the staircase generated by the SLM pixels can lead to unexpected errors for certain steering angles and they optimize the holograms to minimize the difference between the mean slope of the staircase grating and the desired slope (a similar approach is performed in [17]). D. Engström and co-workers [16] also compare the theoretical predictions to experimental results, by measuring the center of mass of the far-field pattern of the steered beam. Noise in the setup allowed them to find agreement between simulations and experiments down to normalized errors of about 2–3% (steering angles of 2–3 μrad), by artificially limiting the performance of the SLM to *N* = 256 pixels and *M* = 32 phase levels. Recently, C. Wang et al. [18] described the appearance of local precision defects due to the quantization in phase modulation with SLMs and presented an optimization strategy using the symmetrical radial sub-aperture coherence algorithm in two dimensions.

Up to now, the studies to analyze and measure beam steering accuracy in optical traps have been scarce, probably due to the difficulty of measuring positioning with the required precision. Trap positioning errors within ∼2 nm have been found in [14] for three holographic traps, for an SLM with *N* = 1080 pixels and *M*∼200 phase values. As we will see later, these errors are mainly due to errors in phase modulation, that is to say, in the conversion from the gray level (or voltage) addressed to the SLM at each pixel and the actual phase added locally to the incoming beam of light. The effects of these mismatches on the look-up-table (LUT) conversion from gray levels to phases have been widely studied in terms of diffraction efficiency [19,20,21], but not in terms of positioning accuracy.

In this paper, we will study, measure and minimize both the LUT and phase quantization effects on single trap positioning, for a phase only SLM with *N* = 600 pixels and *M* = 2^8^ = 256 gray level values. Prior to the optical trapping experiments, the SLM modulation range will be calibrated. Trap positioning will be obtained indirectly by tracking trapped polystyrene microbeads, allowing unprecedented sub-nanometer holographic trap displacement measurements, corresponding to steering angles of the order of 0.1 μrad.

## 2. Materials and Methods

Our HOT set-up is built around a commercial, inverted microscope (Nikon Eclipse TE2000-U, Appendix A). A linearly polarized continuous-wave laser (IPG YLM-5-1064-LP, TEM_00_, *λ* = 1064 nm) is expanded through telescope 1 (*f*_1_ = 30 mm, *f*_2_ = 100 mm) and modulated by a parallel nematic, liquid crystal on silicon (LCoS) reflective SLM (Hamamatsu X10468-03, 800 × 600 pixels, *p* = 20 μm pixel size). This is an analog-addressed SLM with 256 gray levels, controlled by 8-bit digital-video-interface (DVI) signals [22]. A half-wave plate (HWP 1) placed before a polarizing beam splitter (BS) rotated the incoming beam polarization to control the power transmitted through the BS. The extraordinary refractive index of the liquid crystal SLM lies on the horizontal plane, thereby the incident beam is split into a modulated beam with horizontal linear polarization and a non-modulated beam with vertical linear polarization. A second half-wave plate (HWP 2) was used to rotate the beam polarization to 45° so that the two components had the same power. The beam enters the rear fluorescence port of the microscope and is reflected by an IR shortpass dichroic mirror (IR-DM) after being resized by telescope 2 (*f*_3_ = 150 mm, *f*_4_ = 100 mm) to fit the entrance pupil of a water immersion microscope objective lens (Nikon Plan Apo, 60x, NA = 1.2). Its focal length is *f* = *f*_tube_/*m* = 3.33 mm [23], where *f*_tube_ = 200 mm is the tube lens focal length and *m* = 60 is the objective magnification. A condenser illuminates the sample from above under Köhler configuration and the brightfield image is recorded with a CCD camera (QICAM, 1392 × 1040 pixels, 4.65 μm pixel size). Microchambers were built by gluing two cover-glasses with 100 μm thick double-sided scotch tape with a hole (1 × 1 cm) containing 1 μm polystyrene spheres (Sigma-Aldrich) highly diluted in distilled water.

As described in the following lines, we compensated drift motion by subtracting the position of the on-axis, non-modulated trap, to the steered trap, as shown in Figure 1a. Trap positioning was thereby analyzed by moving the holographic trap at steps of different size for 4 s, such that an average value and standard deviation was determined, as indicated in Figure 1d.

## 3. Results

In [24], F. Marsà et al. showed that the low-frequency position drift of a trapped microsphere could be compensated in a dual-trap, dumbbell configuration, which was obtained by splitting the laser beam into two orthogonal, linearly polarized components. A dynamic trap was created from the polarization component modulated by the parallel nematic SLM, while a static, reference trap arose from the non-modulated component. Under this scheme, relative drift motion was reduced to a few Angstroms. Interestingly, crosstalk from the two polarization components was determined below 1%, which we assume does not affect trapping dynamics of the non-modulated trap. Likewise, pointing stability of HOT was assessed below 1 nm for our analog-addressed SLM [22]. Taken together, this allowed us to investigate the sub-nanometer positioning performance of our HOT set-up. In Figure 1a, we show the position traces of the beads in the on-axis, non-modulated trap (bead 1) and in the modulated trap (bead 2) fixed at a permanent position *x*_2_ = 4 μm. As can be observed, both beads undergo perfectly parallel, low-frequency drift motion, which reduces to sub-nanometer fluctuations for the relative position.

### 3.1. Trap Steering in HOT

Under the pixelated performance of an SLM, an optical trap focuses at a distance *d* by displaying the following phase profile onto the liquid crystal plane (Appendix A):(1)$$\phi_{j}^{ideal}\left(d\right)=\frac{2\pi }{\lambda f}\frac{f_{4}}{f_{3}}dx_{j}+\phi_{0}\;;\;\;\;j=1,\ldots ,N$$
where *x_j_* = *p j* is the pixel position and *N* is the number of pixels of the SLM along the steering direction [25]. One immediate result of hologram discretization is the decrease in trap steering efficiency. In our set-up [10], this was confirmed to follow a *sinc^2^* curve, suggesting no additional efficiency loss besides that resulting from the finite size of the pixel. For the range of distances studied in this paper, it is worth noting that the power at the modulated trap remains constant within 2%. We assume this to have a minor impact in bead 1 effective positioning.

Because SLMs can only apply a finite phase delay onto the wavefront, linear phase profiles such as that in Equation (1) become periodic phase gratings with phase values ideally ranging from 0 to 2π. Moreover, the accessible phase values are quantized, typically to 256 for an 8-bit SLM controller (Appendix A):(2)$$\phi_{j}^{actual}\left(d\right)=\mathrm{round}\left[\phi_{j}^{ideal}\frac{M}{2\pi }\right]\frac{2\pi }{M}\;;\;\;j=1,\;\ldots ,\;N$$
where *M* is the so-called gray level range. Here, a gray level *g* applies a phase delay of *φ_g_* = 2π*g*/*M*. In the following sections, we will present an ellipsometry-based strategy for determining the gray level range, *M* and will analyze how the finite phase and phase quantization alter trap positioning in holographic optical tweezers.

### 3.2. Laser Power Effects

We first characterized the phase modulation by sandwiching the SLM between crossed polarizers and measuring the transmitted power with a power meter (Thorlabs), as described in [22]. As the gray level displayed was changed from 0 to 255, the transmitted intensity followed a *sin*^2^ curve from a minimum value (for no phase delay) to a maximum value (for π phase delay). To our surprise, the gray level range, *M*, fell below 256 and increased with laser power, as shown in Figure 1b. We repeated the cycle at every 20 s to assess possible time-dependent, long-term effects and found that *M* exhibited a steep transient during the first 4–5 min—especially for higher powers—and later stabilized for long periods of time, showing plausible heating of the liquid crystal display by the IR laser. This increase in the gray level range is the result of a decrease in the liquid crystal birefringence caused by temperature [26]. In turn, this translated into evident changes in trap positioning for increasing laser powers (Figure 1c) and a noticeable positioning transient over a similar timescale (Figure 1a).

We hypothesized that such instability in trap positioning was due to modulating the beam with an inaccurate LUT, i.e., with an incorrect *M* value. If the gray level range differs from the real 2π phase value (*g_M_* ≠ 2π), a discontinuous phase jump will be produced for each period. Incorrect phase modulation has been widely determined to decrease trap steering efficiency [19,25,27], but it likewise constitutes a source of inaccuracy in trap positioning.

### 3.3. Positioning Deviation Due to Non-Ideal LUT

We simulated trap positioning with a simplified, Fourier optics model, that captures such LUT-originated positioning deviation and suggests an experimental strategy to correct for it. Let us consider an incorrect gray level range, *M*’ = *aM*, where *a* is a scaling factor and *M* the real gray level range (see Figure 2a). The periodic phase grating profile at the SLM plane can be described as: (3)$$U\left(x\right)=A\cdot rect\left(\frac{x-x0}{L}\right)\left[\overset{\infty }{\underset{n=-\infty }{\sum }}\delta \left(x-nT\right)*\left(e^{i\frac{2\pi a}{T}x}\cdot rect\;\left[\frac{x-\frac{T}{2}}{T}\right]\right)\right]$$
where *T* is the period of the grating and *L* and *x*_0_ are the length and center position of a window defining the finite size of the SLM, respectively. In Equation (3), we are omitting both discretization due to finite pixel size and quantization of the phase. When focused through the objective lens (focal length f), the optical field is described by the optical Fourier transform. In our case, $u=\frac{x\prime }{\lambda f}\frac{f_{4}}{f_{3}}$ is the spatial frequency and *x*’ the coordinate at the trapping plane. We obtained the following summation of diffractive orders, modulated by a *sinc* function: (4)$${\left|\tilde{U}\left(u\right)\right|}^{2}\propto {\left|\overset{\infty }{\underset{n=-\infty }{\sum }}sinc\left(n-a\right)e^{i\pi \left(\frac{2x_{0}-1}{T}\right)n}sinc\left[L\left(u-\frac{n}{T}\right)\right]\right|}^{2}$$

Each diffractive order appears at $d_{n}=n\frac{\lambda f}{T}\frac{f_{3}}{f_{4}}$ [25] and is modulated by the factor $sinc\left(n-a\right)e^{i\pi \left(\frac{2x_{0}-1}{T}\right)n}$. For the ideal, perfect phase modulation, i.e., *a* = 1, this *sinc* factor vanishes for every *n* other than 1, yielding an optical trap at *d*_1_ with 100% of the power. Note that the primary effect of *a* ≠ 1 is thereby a decrease in trap steering efficiency, as depicted in Figure 2b. In a similar analysis based on Fourier Optics [27], J. Albero et al. described that *a* = 0.5 leads to two optical taps at $d_{n=0}$ and $d_{n=1}$, with 40.5% efficiency each, as ${\left|sinc\left(0.5\right)\right|}^{2}=0.405$. 

However, from a closer observation of the diffracted optical field, as shown in Figure 2b for an exaggerated case with *a* = 0.6, we can see that the peak position also deviates from the ideal, *x*_trap_ = *d*_1._ The reason for this deviation is a little crosstalk from secondary peaks into the trap created at *d*_1_. When moving the optical trap continuously, the simulated position deviates periodically from the target (Figure 2c), with a typical, so-called L-distance, given by:(5)$$d_{L}=\frac{\lambda f}{L}\frac{f_{3}}{f_{4}}$$

In our set-up, *d_L_* = 0.443 μm. It is worth observing that, for smaller steering distances, trap positioning is not affected by LUT inaccuracy, meaning that such source of inaccuracy is visible on a large scale compared to the sub-nanometer steps that will be analyzed later. As shown in Figure 2c, for the case *a* = 0.9, trap positioning can deviate around 5–10 μm from the target value.

Interestingly, we noted that *x*_trap_ is affected by lateral shifts in the hologram, expressed by parameter *x*_0_ in Equation (4). For convenience, we can apply such shifts by adding a phase, *φ*_0_, to the hologram. In Figure 2c, the trap is steered from *x*_trap_ = 1 μm to *x*_trap_ = 5 μm for different values of *φ*_0_, evidencing the change in the trap position. We will use this property to optimize the positioning accuracy and approach *x*_trap_ = *d*_1_.

This long-range inaccuracy was also observed in experimental positioning, as shown in Figure 3a for an artificial reduction of phase modulation to *a* = 0.9 (*M*’ = 205), similar to Figure 2c. We then used the gray level range that was calibrated from Figure 1b, i.e., *M* = 228 for *P* = 1W. When moving the optical trap over 1 μm (Figure 3b), we clearly observed positioning deviations of up to ± 2 nm, resulting in a root-mean-square (RMS) of 1.7 nm. The periodic undulation exhibited in the profile simulated in Figure 2c and measured experimentally for *M*’ = 205 in Figure 3a, however, was no longer visible. Since we were using the calibrated *M* value, we expect that errors in experimentally measured spatial positioning could also arise from other features, such as a spatially varying SLM LUT [19], pixel cross-talk [21] or even optical aberrations, which were not included in the simulation. Nevertheless, we could improve optical trap positioning with the addition of optimized phase values, *φ*_0_, to each of the computed holograms. The optimal values were found by measuring the changes in the experimental trap position around the target position as *φ*_0_ was varied, as illustrated in Figure 3c. Figure 3d shows trap positioning after correction, which was brought down to sub-nm accuracy, yielding an RMS value of 0.34 nm.

### 3.4. Positioning Deviation Due to Phase Quantization

In [16], D. Engström et al. realized that steering angles deviated from the ideal due to phase quantization (Appendix A). By using *φ*_0_ as a free parameter, they optimized the positioning of the laser spot onto a CCD camera. Here, we measured the same positioning alteration on an optically trapped microsphere and used the same strategy to correct for trap positioning.

We observed that deviations originated from phase quantization appear dramatically around positions defined by perfect phase slopes, i.e., those defined by an integer number of gray levels per pixel. In terms of phase values, this can be expressed as $\phi_{j}=n\frac{2\pi }{M}j$. The so-called M-positions are defined as:


(6)$$d_{M}^{\left(n\right)}=n\frac{\lambda f}{Mp}\frac{f_{3}}{f_{4}}$$


In our case, $d_{228}^{\left(n\right)}=1.17n$ (μm).

In addition, as reported in [16] and [18], such local precision defects are periodically reproduced along the steering direction. Without the loss of generality, we analyzed the effect at $d_{288}^{\left(4\right)}=4.662$ μm, such that the two microbeads were separated enough for accurate video tracking. In Figure 4a, we show the trajectory followed by a trapped bead, 20 nm forward at steps of 0.2 nm, from $d_{288}^{\left(4\right)}$. We distinguished two domains according to the trap positioning behavior. For the first 5 nm, trap position deviated up to 2 nm and yielded an RMS of 1.1 nm. Beyond 5 nm, positioning was not affected by phase quantization, confining the positioning accuracy within an RMS of 0.21 nm around a straight line.

In Figure 4b, we show the variation in trap positioning obtained by the addition of a small phase into the hologram computation. It is worth mentioning that the phase offset, *φ*_0_, must now correspond to a gray value between 0 and 1, in order to apply a change in the actual phase displayed after quantization (Appendix A). For two particular positions (Figure 4b(i,ii)), note that this can be used to approach the position target. We repeated this procedure for all the trap positions in the affected domain (A), which allowed us to reduce the positioning error from an RMS of 1.1 nm down to 0.29 nm, similar to that along the non-affected domain (NA).

## 4. Discussion and Conclusions

In this paper, we describe two sources of positioning inaccuracy in holographic optical traps and proposed a strategy to achieve sub-nanometer accuracy. First, we noted that the gray level range of our SLM, i.e., the number of gray levels available from 0 to 2π, increases with laser power. A mismatch between the real gray level range *M* and that used for hologram calculation leads to errors of about 2 nm. This is observed when steering the trap over a long range, defined by the *L*-distance, which is *d_L_* ~ 0.443 μm for our HOT set-up. Positioning can be corrected for by shifting the hologram, which we implemented by the addition of a phase offset, *φ*_0_. The trap deviation profile simulated by Fourier Optics shows periodicity over *d_L_*, which we observed experimentally by decreasing the SLM phase modulation range down to *a* = 0.9. However, when using *M*’ = *M* = 228, i.e., *a* = 1, trap position deviations did not follow this trend and were likely affected by smaller features, such as optical aberrations or a spatially-varying LUT.

Over a smaller, nanometer-scale range, positioning deviations of up to 2 nm became evident around certain trap locations, the so-called *M*-positions. Such deviation is originated by an incorrect phase slope, which arises from phase quantization in digital holography. Similar to long-range deviations over distances comparable to *d_L_*, the phase slopes can also be corrected by the addition of a phase offset, *φ*_0_. Beyond 5 nm from *M*-positions, sub-nanometer accuracy in trap positioning is achieved with no need for hologram correction.

The sources of inaccuracy reported in this paper are likely to affect trap positioning in multiple HOTs [14]. However, a single correction parameter, such as *φ*_0_, will not suffice for independent correction for the positioning of each trap. Several approaches to correct for trap steering efficiency in multiple HOT have been reported, with special insight into spatially varying LUT [9,19]. In our opinion, similar corrections can be used to improve trap positioning accuracy in multiple HOTs. The temporal stability of our correction approach is different for the two regimes discussed here. As shown in Figure 1, LUT evolution caused by laser power induces a positioning transient drift at the minute scale. Positioning correction must therefore be carried out once the LUT stabilizes. Since we observed it to be noticeably dependent on laser power and other experimental parameters, it is worth considering an in situ optimization right before the experiment. In addition, we recommend SLM cooling strategies to avoid undesired changes in phase modulation efficiency [28].

In contrast, we observed that nanometer-scale deviations around local precision defects due to phase quantization were stable in time. Importantly, while the absolute position could be drifted by LUT-induced positioning deviations, quantization-induced errors around $d_{M}^{\left(n\right)}$ were robustly corrected by the optimization of the phase ramp slope by the addition of *φ*_0_. We think this is the first time that such small-scale effects on trap positioning could be experimentally measured and corrected for and make holographic optical tweezers a suitable tool for high precision experiments.

## Figures and Tables

**Figure 1 micromachines-12-00559-f001:**
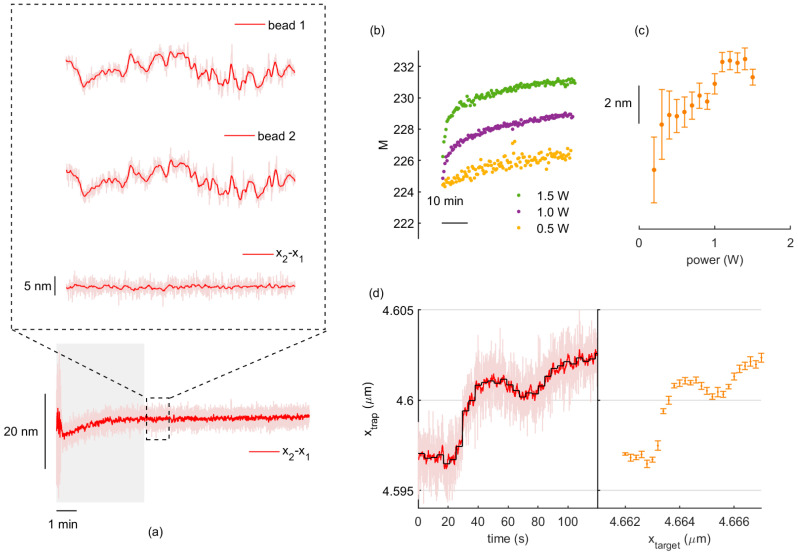
Drift cancellation and positioning dependence on laser power. (**a**) Position traces from video tracking on two optically trapped beads undergoing parallel drift motion. (**b**) Evolution of SLM calibration over time for three different laser powers. (**c**) Trap positioning variations originated for increasing laser powers. Each point and error bar are the mean and standard deviation of the relative trap position over 4 s. (**d**) Left: relative position trace for holographic trap steps of 0.2 nm at every 4 s, around $d_{M=228}^{\left(n\right)}$. In red, the signal is smoothed with a Savitzky–Golay filter of 1 s. In black, the average position is indicated as a staircase. Right: for each step in the left plot, a mean value and standard deviation, *σ*, are obtained.

**Figure 2 micromachines-12-00559-f002:**
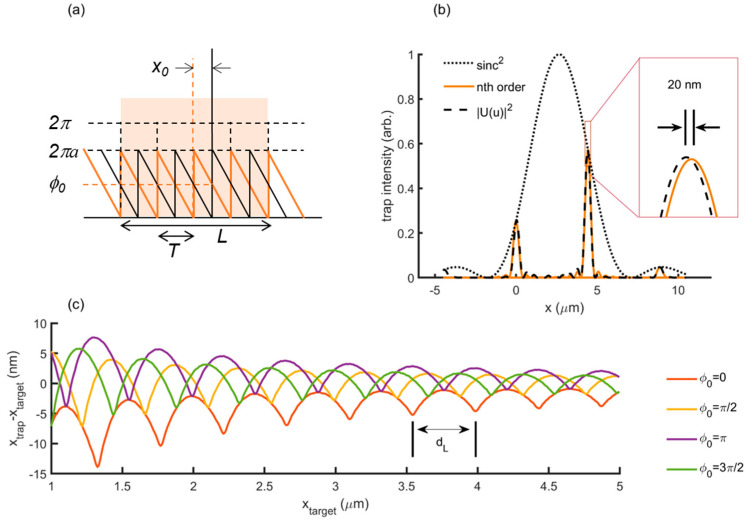
Fourier Optics simulation of trap positioning deviations. (**a**) Periodic phase grating displayed at the SLM plane, expressed as the convolution of phase ramps separated a distance *T* and limited by the SLM window, *L*. Incorrect phase modulation is expressed by factor *a*. Note that a global phase, *φ*_0_, translates into a hologram shift, *x*_0_. (**b**) Intensity profile at the trapping plane for *a* = 0.6. The maximum peak for the diffracted field, ${\left|\tilde{U}\left(u\right)\right|}^{2}$, differs from the ideal position of the first diffraction order, i.e., *x*_trap_ ≠ *d*_1_. Coordinate *x* refers here to the trapping plane. (**c**) Long-range positioning deviation simulated for different phases, *φ*_0_, with *a* = 0.9.

**Figure 3 micromachines-12-00559-f003:**
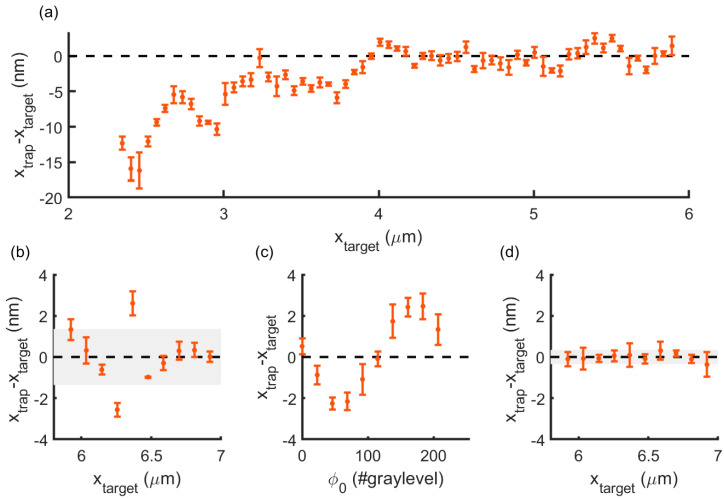
LUT-induced positioning inaccuracy. (**a**) Trap positioning deviation along a displacement of several microns for an exaggerated case of *a* = 0.9 (*M*’ = 205). (**b**) Same for *M*’ = *M* = 228 and a shorter range. RMS: 1.7 nm. (**c**) Variations in trap positioning originated by the addition of *φ*_0_. (**d**) Trap positioning error along the same displacement in (**b**), with each position corrected with *φ*_0_. Data points are mean +/− *σ* over 4 s of relative trap position measurements.

**Figure 4 micromachines-12-00559-f004:**
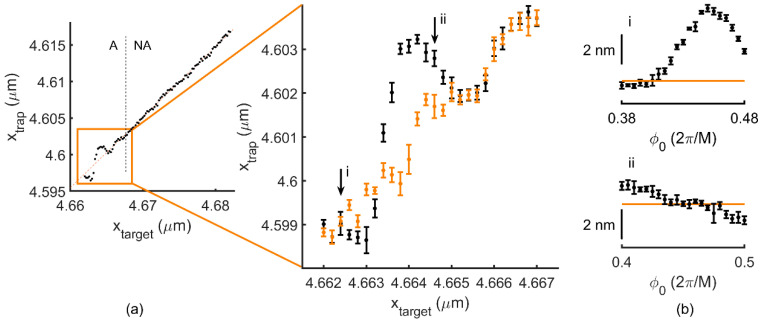
Trap positioning deviations originated from phase quantization. (**a**) Measured trap position versus target position over 20 nm from $d_{228}^{\left(4\right)}=4.662\;$μm. Affected (A) and non-affected (NA) domains are delimited by the dashed line. Inset: Non-corrected (black, RMS = 1.1 nm) and corrected (orange, RMS = 0.29). (**b**) Positioning variation for positions i and ii, indicated in (**a**), for different phases. The orange line indicates the ideal position. Data points are mean +/− *σ* over 4 s of relative trap position measurements.

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
