# Peer review of "Positioning Accuracy in Holographic Optical Traps"

_micromachines, 2021, doi:10.3390/mi12050559_

Round 1
Reviewer 1 Report
The authors present experimentally the positioning issue in holographic optical traps introduced by the limits of the SLM device. And a solution is given as followed. It is interesting for the readers in the community of optical tweezers, who want to know how to precisely manipulate the traps. Before the manuscript is accepted, I have the following questions or recommends as below.
- A model of two traps in this paper are tested and discussed. The results are good. But, is the situation also the same when more traps are created by a holographic way? That is to say, is the report a common solution to holographic optical traps?
- A distance of 4 micrometer between two traps are chosen. Why? Can the distance be smaller or larger? Then, what happens? Here, I want to say that it is better to tell the reader more clearly the choose of the parameters used in the manuscript.
- It is better to add some illustrates for a well understanding of the equations.
- Does the pointing direction of the laser beam move when adjusting the laser power by rotating the optics?
- "a water immersion microscope objective lens (Nikon Plan Apo, 60x, NA=1.2), with a focal length of f = 3.33 mm". Is the focal length is correct?
- The last question, how long time does a calibration data last for a holographic optical trapping system?
Reviewer 2 Report
The present manuscript address positioning inaccuracies for holographic optical trap (HOTs). HOTs are the most flexible and creative tool to create and position optical traps in 3D even with exotic point spread functions. Their application usually requires applying corrections due to aberrations. The present manuscript elegantly describes positioning inaccuracies arising from the discrete nature of SLMs and how to correct for them.
The manuscript is divided into 1) effects due to laser power, 2) non-ideal LUTs, and 3) phase quantization. All three issues are quantified separately and strategies for how to overcome these limitations in the background of high accuracy experiments are presented. Inaccuracies due to laser power affects the maximum level of gray values that the SLM can display, which is measured and then included in the phase algorithm to display the correct LUT. Long and short range inaccuracies due to incorrect LUTs and phase quantization are corrected by introducing a compensating phase shift, which is determined empirically and then applied as a correction.
The authors method achieves evidently a great positing improvement for high precision experiments such as molecular motor studies and thus presents noteworthy novelty. I recommend publication after my points below have been addressed.
- One general point that I have is the reproducibility of the experiments. Are the presented data, means, and standard deviations the result of many repeats of the same experiment? If yes, how many experiments were performed? If not, how reproducible are the position errors with and without compensation for several repeated experiments?
- Line 28: the term ”object-adapted optical trapping” sounds very interesting. However, a quick google scholar search revealed that this terminology is mainly used for traps created by acousto-optic deflectors.
- What is the power and position stability of the fixed trap (trap 1 / bead 1)? This trap is created by using the non-modulated polarization component reflected by the SLM. However, no polarization optic is perfect. I could imagine that typical extinction ratios of 1:100 and in addition crosstalk from manipulating one polarization to the other could yield similar small positioning inaccuracies on the Angstrom range as addressed in the manuscript.
- Line 89: What does “bead 2 steered at x2 = 4um” mean? My understanding is that trap 2 is maintained at x = 4um while trap 1 remains at x = 0. Is this correct? The term steered implies a motion as opposed to remaining the position of the trap….
- Line 112: I would like to see the transmitted intensity dependence as a plot, maybe in the Supplement.
- Line 114: How is the value of M measured? It is interesting that heating of the SLM affects the maximum levels of gray values that the SLM can display and not the accuracy at which a specific value is displayed. This might be a question that is too technical, but is it clear why this is? It seems a bit counterintuitive that the max M increases with laser power, naively I would expect it to decrease…
- Fig 1b): How does this effect affect the position accuracy of the modulated trap as a function of time? How does this heating effect affect the non-modulated polarization reflected by the SLM to create the fixed trap?
- Fig 1b,c): This laser power dependence basically means that for high precession experiments, the laser power and thus the trap stiffness can not be altered during an experiment. Is it possible to cool the SLM more to avoid this effect?
- Line 132: The period T of the grating represents the finite pixel size, does it not?
- Lines 130 – 139: I am a bit confused as to what values n can have. From lines 133, 134 it seems n starts at n = 1, but in line 139 it seems n starts at n = 0. This is important for understanding the sinc function in line 136 and Eq. 4.
- 3a): Why was the range of x values chosen to be limited to 6 – 7 um? The simulations in Fig. 2 are from 1 – 5 um. I would like to see this larger range and if the predicted periodicity is visible over this range.
- 3): Data points are the mean and standard deviation of what? How often was this experiment repeated? What about positions x < 4.66 um? I expect those values to be as accurate as values x > 4.67?
- S2): I think this Fig is actually very helpful. If the spatial constraints of the manuscript allow, I would encourage the authors to elevate this supplemental figure to a main text figure, at their discretion.
- Lines 201-207: How was the additional phase determined? Imperially and similar to the method described in lines 171-174?
- The current manuscript addresses positioning inaccuracies only with regard to one dimensional position changes. The great benefit of HOTs is though the 3D positioning ability and creating and manipulation of multiple optical traps simultaneously. In theory, could the same algorithm work in 3D and for multiple traps as well?
Round 2
Reviewer 2 Report
The authors addressed all of my previous comments and questions and I recommend publication in the current form.